# A Real Time Arabic Sign Language Alphabets (ArSLA) Recognition Model Using Deep Learning Architecture

**Zaran Alsaadi [1], Easa Alshamani [1], Mohammed Alrehaili [1], Abdulmajeed Ayesh D. Alrashdi [1], Saleh Albelwi [1,2] and Abdelrahman Osman Elfaki [1,*]**

[1] Faculty of Computing and Information Technology, University of Tabuk, Tabuk 71491, Saudi Arabia; 421009914@stu.ut.edu.sa (Z.A.); 421009814@stu.ut.edu.sa (E.A.); 421009808@stu.ut.edu.sa (M.A.); 421009856@stu.ut.edu.sa (A.A.D.A.); sbalawi@ut.edu.sa (S.A.)

[2] Industrial Innovation and Robotics Center, University of Tabuk, Tabuk 71491, Saudi Arabia

[*] Correspondence: a.elfaki@ut.edu.sa

**Abstract:** Currently, treating sign language issues and producing high quality solutions has attracted researchers and practitioners' attention due to the considerable prevalence of hearing disabilities around the world. The literature shows that Arabic Sign Language (ArSL) is one of the most popular sign languages due to its rate of use. ArSL is categorized into two groups: The first group is ArSL, where words are represented by signs, i.e., pictures. The second group is ArSl alphabetic (ArSLA), where each Arabic letter is represented by a sign. This paper introduces a real time ArSLA recognition model using deep learning architecture. As a methodology, the proceeding steps were followed. First, a trusted scientific ArSLA dataset was located. Second, the best deep learning architectures were chosen by investigating related works. Third, an experiment was conducted to test the previously selected deep learning architectures. Fourth, the deep learning architecture was selected based on extracted results. Finally, a real time recognition system was developed. The results of the experiment show that the AlexNet architecture is the best due to its high accuracy rate. The model was developed based on AlexNet architecture and successfully tested at real time with a 94.81% accuracy rate.

**Keywords:** deep learning; Arabic Sign Language alphabetic; AlexNet architecture; transfer learning; data augmentation

## 1. Introduction and Motivation

Sign language is a method used by people with hearing impairments to interact with others. Therefore, research and development in the field of sign language recognition has attracted scientists, researchers, and engineers to develop software to facilitate the process of communicating with people with hearing disabilities [1]. Sign language is a form of communication that uses well-known signs or body motions to convey meaning. There are many hearing impaired people who are unable to write or read a language, as well. Hence, building a sign language translation or, in other words, a sign language recognition (SLR) system can be extremely beneficial to their lives. The SLR system is in high demand due to its capacity to bridge the gap between the hearing impairmed community and the rest of the world. It is one of the most important areas of computational research that deals with real life issues. According to the World Health Organization, Fact Sheet (2022), over 5% of the world's population suffer from hearing impairments. It is also estimated that this number will increase to above 700 million people by 2050.

Arabic alphabets are used by the population of Arab countries, which amounts to about 1 billion people, or 14% of the world's population (World Health Organization, 2015). In addition to the population of Arab countries, many Asian and African populations use Arabic alphabets in their languages or dialects, such as Persian, Malay (Jawi), Uyghur, Kurdish, Punjabi, Sindhi, Balti, Balochi, Pashto, Lurish, Urdu, Kashmiri, Rohingya, Somali

and Mandinka, among others [2]. Therefore, Arabic alphabets are used by almost a quarter of the world's population, clearly illustrating their significance.

According to the law of multiple proportions, 5% of the populations that use Arabic alphabetics face difficulties with hearing impairments. This is considered to be a significant number. This justifies the importance of Arabic Sign Language Alphabets (ArSLA). Arabic Sign Language Alphabets (ArSLA) are an illustration of Arabic letters in sign language shapes. Hence, ArSLA is used by the hearing-impaired community to overcome the obstacle of dealing with traditional Arabic letters. This enables them to be involved in the traditional educational and pedagogic process [3].

Arabic sign language is divided into two parts: the first part is a complete language where each word is represented by a sign (for instance, the word father is represented by a sign). Different Arab countries have their own Arabic sign languages, such as Egyptian sign language, or Saudi sign language. The first part is known as Arabic Sign Language (ArSL). In the second part, each letter in the Arabic alphabet is represented by a special sign known as Arabic sign Language Alphabets (ArSLA). Due to the formerly explained importance of Arabic sign language, the challenge of developing Arabic sign language recognition systems has garnered the attention of researchers and practitioners. As a result, the literature suggests many solutions for both ArSL and ArSLA.

Transfer learning [4] has been proposed as a solution to overcome this challenge, a technique in which the model is trained on a large training set. The results of this training are then treated as the starting point for the target task. Transfer learning has proved to be successful in fields such as language processing and computer vision. Data augmentation [5] is another technique found to be effective in alleviating overfitting and improving overall performance. This method increases the training set's size by performing geometric and color transformations such as rotation, resizing, cropping, and adding noise to or blurring the image, etc. In this work, we did not use either transfer learning or data augmentation when training our CNN model. Instead, we utilized the ArSLA dataset which is suitable for testing and training. It uses around 1000 images for each letter to train the CNN model.

This paper presents a real time Arabic Sign Language Alphabets (ArSLA) recognition model using deep learning techniques. As previously discusses, we focused on ArSLA due to its popularity. The main target was to develop a solution that can be accessible for everyone.

## 2. Related Works

This section explores related works that were investigated and analyzed with the aim of discovering and highlighting any research gaps. The methodology used for collecting related works was to select recent research that utilized sign language recognition methods covering the previous 10-year period. As selection criteria, research papers that addressed issues with ArSL, or ArSLA with applicable solutions were chosen. The research papers that presented inapplicable or reiterated solutions were neglected.

According to [6], the recognition systems of sign languages could be categorized into two groups: glove-based systems and vision-based systems. The first group is based on hardware devices which consist of special sensors that can be packaged in different shapes that should be suitable to be used by hand (since sign languages are characterized by hands). The second group is based on image processing techniques and algorithms which leads to using only the camera. Despite the promising achievements in the first group, the second group could still be considered the best choice as the only hardware device it requires is a camera which available in almost any modern computer. In the literature, the first group is titled as sensors-based solutions, while the second group is titled as image-based solutions. In this paper, we follow the concepts of the second group. In the following, selected papers from related works are discussed and analyzed.

Halawani and Zaitun [7] developed a system for converting common Arabic spoken words to ArSL using a speech recognition engine. This work suggests using data gloves

for measuring sign language motion. It is not clear how the common words used in the system were collected and validated. Mohandes el al. [8] developed a multilevel system for ArSL recognition. In the first level, the leap motion controller is used for tracking and detecting hand motion. Level one is used for images acquisition with the aim of creating a dataset. The second level is a preprocessing of collected images. The third level is a feature extraction process. The final level is a classification model. The performance of such systems is dependent on the degree of accuracy in defining image features. ElBadawy et al. [9] used a 3D Convolutional Neural Network (CNN) to develop an ArSL recognition system based on 25 sign pictures. The results of this research achieved 85% accuracy. Alzohairi et al. [10] developed an ArSLA recognition system using a Support Vector Machine (SVM). The SVM has been implemented as one versus all SVM that extract the histograms of Oriented Gradients (HOG) descriptor. The accuracy of this system did not exceed 63%.

Ibrahim et al. [11] proposed an automatic visual Sign Language Recognition System (SLRS) that converts solitary Arabic words into text. This proposal is limited to the 30 isolated words used in the daily school life of hearing-challenged children. A proposed skin-blob tracking technique is used to identify and track the hands.

Deriche et al. [12] designed an ArSL recognition system based on dual Leap Motion Controllers (LMC). The optimum geometric features were collected from both front and side LMCs. The classification was developed on aBayesian approach. The system was validated on one hundred developed signs. The accuracy of this system was not mentioned. Hassan et al. [13] conducted experiments for comparing the two methods of ArSL recognition which are: the k-Nearest Neighbor and Hidden Markov Models. The experiments were conducted based on a dataset collected by using sensor gloves and another dataset collected by a motion tracker. The results showed a similar classification accuracy. This means that the acquisition methods do not affect classification accuracy since a dataset collected them correctly. Gangrade and Bharti [14] used multi-layered random forest (MLRF) for recognizing static gestures from depth data provided by Microsoft's Kinect sensor. The method in [14] was validated by synthetic data, a publicly available dataset of 24 American Sign Language (ASL) signals.

Kamruzzaman [15] proposed an ArSLA recognition system aimed at translating signs to Arabic speech. The classifier was developed based on a Convolutional Neural Network (CNN). The CNN architecture that was used in this model is not clear. The work in [16] developed a comparison survey to study the performance of ArSLAN classifiers. The selected classifiers were CNN, RNN, MLP, LDA, HMM, ANN, SVM, and KNN. The works in [14, 1] used deep convolutional networks to recognize letters and digits in American Sign Language. Missing signs (special letters, such as the space sign or gap between words) in the corpus are a source of errors in these models. In addition, the used CNN architectures are not clear.

The discussion of related works reflects the following issues which represent the research gap:

(1) There are many works suggesting the use of a special hardware device to read hand signs which is considered an extra burden and an added expense. These solutions will not be available to everyone.

(2) Related works that have been developed based on machine learning, or image processing algorithms, are dependent on implementing feature extraction techniques. Hence, the quality of extracted results is completely subject to the selected features which could possibly be an imperfect selection.

(3) Related works that have been developed based on CNN neglect well known CNN architectures. Developing a solution based on Ad hoc CNN architecture is doubted on solution credibility, as new CNN architecture should be tested and validated in different environments and situations.

(4) The works that dealt with ArSL suffer from scalability issues. Not all words can be covered due to the vast amount of words used in sign language.

In the following sections, explanation of how the proposed model has overcome the above shortcomings has been presented in detail.

## 3. Methodology

In this section, the steps that have been followed to achieve the proposed model are presented. As mentioned in the related works, ArSLA recognition could be achieved based on machine learning, or deep learning, i.e., CNN techniques. In this paper, we have chosen CNN to develop the recognition model due to its advantage over machine learning. Contrary to machine learning, the CNN model tackles the feature extraction by itself. The basic design principles for a CNN are to construct an architecture and a learning algorithm in such a way that the number of parameters is reduced but the compression and prediction capacity of the learning algorithms is not compromised. CNN layers and nonlinear activations are commonly used after the linear math procedure of convolution. Many times, in the architecture, local connections between pixels are utilized. The introduction of a local receptive field allows for the extraction of various feature elements. Hidden layers placed in-between fully connected layers, can detect a higher degree of complexity. CNN is more effective than machine learning recognition, reconstruction, and classification because of functions of sparse connectivity between subsequent layers, the parameter sharing of weights between neighboring pixels, and similar representations.

Figure 1 shows the frame of methodology for developing the proposed model. This framework consists of five steps: the search for a suitable standard ArSLA dataset, the search for suitable CNN models, selection of the best CNN model, the development of a real time recognition system, and validation of the developed real time recognition system. In the following, each step is presented and discussed.

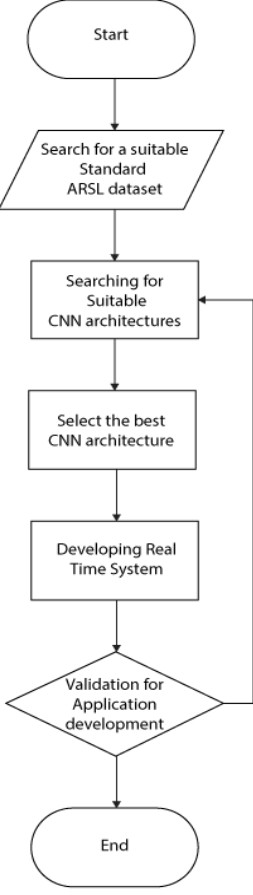

**Figure 1.** Frame of methodology for developing the proposed model.

### 3.1. Search for a Suitable Standard ArSLA Dataset

In this step, we searched for a suitable standard ArSLA dataset. If no suitable dataset was found, there were no other options but to create our own dataset. Figure 2 shows the dataset selection strategy.

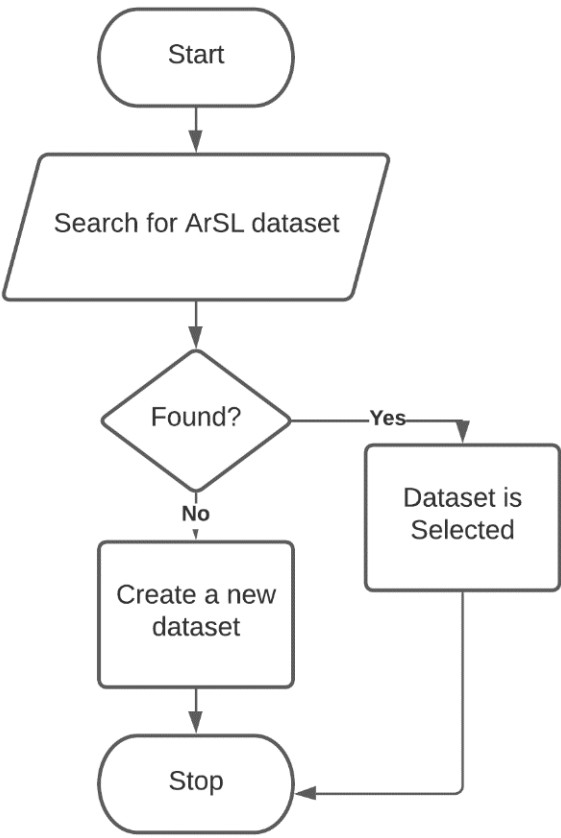

**Figure 2.** The dataset selection strategy.

The targeted dataset should contain three features which are:

(1)  Dataset for ArSLA. This means the dataset should contain all Arabic alphabetics letters.
(2)  The dataset should consist of static images.
(3)  The dataset should be a standard dataset which means the selected dataset has been involved in public research with published results.

We have chosen the ArSLA dataset that was published in [6]. This dataset has the advantages of being fully tagged, made publicly available, consisting of 54,049 images in greyscale jpg formats with a resolution of 64 × 64, and representing the 32 Arabic letters. Figure 3 shows the selected ArSLA dataset.

Figure 3 shows 32 basic Arabic signs and alphabets that contains 54,049 photos of ArSLA demonstrated by more than 40 people. The number of photos per class varies depending on the class. A sample graphic of all Arabic Language Signs is also included. Based on the image file name, the CSV file contains the label of each related Arabic Sign Language Image.

### 3.2. Search for Suitable CNN Architectures

As it is known in computer science, deep learning or CNN is the best technique that can be used for a recognition system. According to [1] the previous most-used deep learning trained architectures are AlexNet [17], VGG-16 [18] and ResNet50 [19], EfficientNet [20] among others.

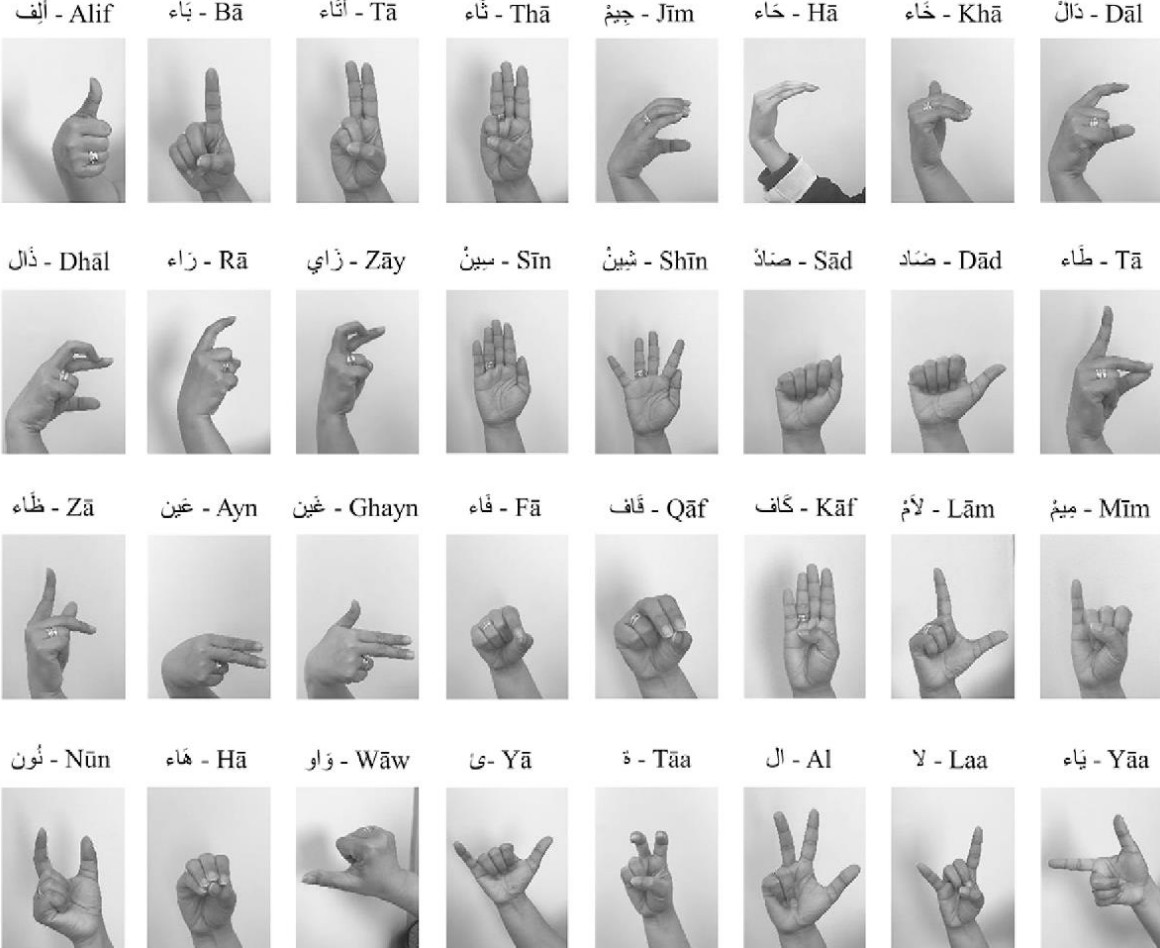

**Figure 3.** The selected ArSLA dataset.

CNN can be fed raw image pixel values rather than pre-processed feature vectors, unlike standard machine learning applications. Figure 4 shows the general architecture of CNN [21]. CNN's usual architecture is made up of layers of computing units (gates), which are:

1.  Convolutional Layers: A grid that provides input to each gate. Each gate's weights are connected so that each gate recognizes the same feature. There are various sets of gates similar to this, organized in multiple channels (layers) to learn different aspects.
2.  Pooling Layers: This works as a down–sampling layer by reducing the number of gates. Each of the "$k \times k$" input grid gates are usually reduced to a single cell/gate by choosing the maximum input value or the average of all inputs. The layer is scanned with a tiny $k$ grid and a stride is chosen so that the grid covers the layer without overlapping.
3.  Fully linked Layers: Each gate's output is connected to the input of the next layer's gate. (Also referred to as auto encoder levels). These transform a vectorized version of the input into a normalized vectorized output. The output vector is a set of probabilities that serve as the classification signature.
4.  Convolution Layers: Consider 1D convolution, suppose the input vector is $f$ and the kernel is $g$ whose length is $m$. The following equation shows the center of kernel shifted and multiplied.

$$(f \times g)(i) = \sum_{j=1}^{m} g(j) f\left( \left\lceil i - j + \frac{m}{2} \right\rceil \right)$$

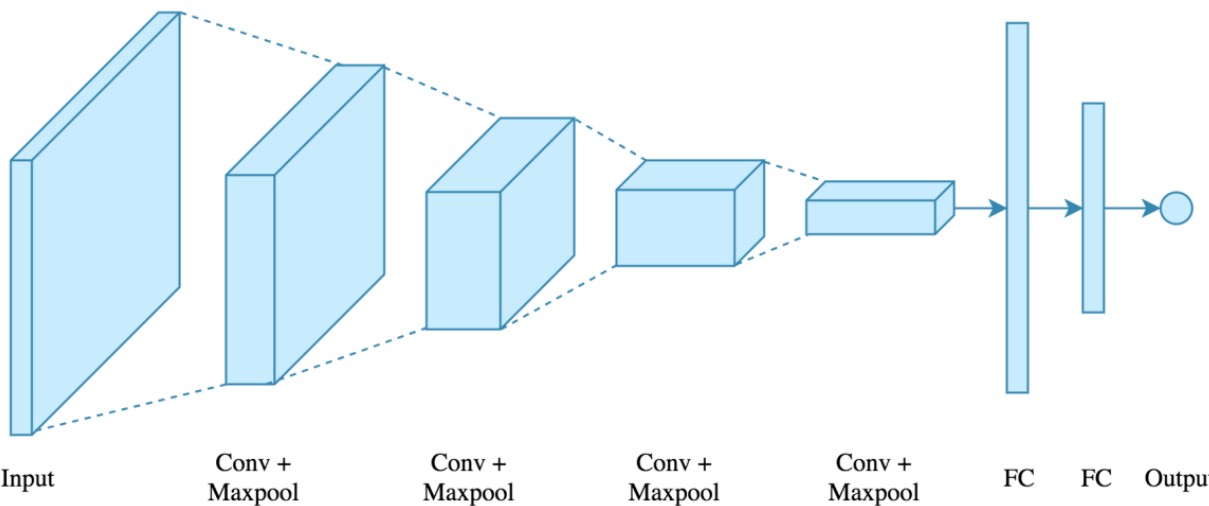

**Figure 4.** General architecture of CNN.

Similarly, one can define the 2D convolution. If the input of 2D convolution is an image $I$ (or, equivalently, a w by h matrix), and the $m \times m$ kernel matrix is denoted as $W$, then this could be noted by the following equation:

$$(I \otimes W)(x, y) = \sum_{j=1}^{m} \sum_{k=1}^{m} W_{j,k} I_{\lceil x-j+m/2 \rceil, \lceil y-k+m/2 \rceil}^{-}$$

Consider next the Convolutional layer in a typical CNN. Suppose the input of the convolutional layer has the dimension $H \times W \times C$, then the convolutional layers can be taken as a set of $C$ parallel, or stacked matrix feature maps, formed by convolving different sized matrix kernels (feature detectors) over the input, and projecting element wise the accumulated dot products. If the chosen convolutional kernel is $k_1 \times k_2 \times C$ and with a stride $Z_s$ (representing the kernel sliding interval), and together with a zero-padding parameter $Z_p$, representing the extent of the zero-border surrounding the image, one controls the size of the resulting feature maps. Then, the dimension of the output of such a convolution layer will be $H_1 \times W_1 \times D_1$, where:

$$(H_1, W_1, D_1) = \left( \frac{H + 2Z_p - k_1}{Z_s + 1}, \frac{W + 2Z_p - k_2}{Z_s + 1}, K_D \right)$$

where additionally $K_D$: =depth size capturing the number of stacked convolutional layers, (=C in this case).

**Activation functions:**

Activation functions define the output of a neuron based on a given set of inputs. Some commonly used activation functions $\sigma()$ with their gradients are as follows (Zanna, and Bolton, 2020) [22]:

ReLU: $\sigma(x) = \begin{cases} 0 & x < 0 \\ x & x \geq 0 \end{cases}$

Softmax: $\sigma(x_j) = \frac{e^{x_j}}{\sum_{k=1}^{d} e^{x_k}}$

5. Pooling Layers: Pooling layers are down-sampling layers combining the output of layers to a single neuron. If we denote $k$ as the kernel size (now assume kernel is squared), $D_n$ as number of kernel windows, and $Z_s$ as stride to develop pooling layers, then the output dimension of the pooling layer will be (suppose we have $H_1 \times W_1 \times D_1$ input) as has been denoted be the following equation:

$$\text{where } (H_1, W_1, D_1) = \left( \frac{H_1 - k}{Z_s} + 1, \frac{W_1 - k}{Z_s} + 1, D_n \right)$$

With a pooling type that can be:

Max-pooling
Average Pooling
L2 norm Pooling

6. Fully Connected Dense Layers: After the pooling layers, pixels of pooling layers are stretched to a single column vector. These vectorized and concatenated data points are fed into dense layers, known as fully connected layers for the classification. In some cases, the output layer of a deep network uses a soft max procedure. Similar to logistic regression: Given n vectors $\{x_1, x_2, \ldots, x_n\}$ with labels $\{l_1, l_2, \ldots, l_n\}$, where $l_i \in \{0, 1\}$ (as a binary classification task). With a weight vector w one can define by the following equation:

$$\text{Prob}(l = 1 \mid x) = \sigma(w^T x) := \frac{1}{1 + e^{-w^T x}}$$

where $\sigma$ represents sigmoid function.

In the following, the selected four CNN models have been presented.

### 3.2.1. AlexNet

AlexNet was the first convolutional network to employ the graphics processing unit (GPU) to improve performance. AlexNet has five convolutional layers, three max-pooling layers, two normalization layers, two fully connected layers, and one SoftMax layer in its design. Convolutional filters and a nonlinear activation function ReLU are used in each convolutional layer. Max pooling is conducted using the pooling layers. Due to the presence of fully connected layers, the input size is fixed. The input size is usually stated as 224 × 224 × 3, however due to padding, it can add up to 227 × 227 × 3. The neural network has 60 million parameters in total. Max Pool is used to down-sample an image or a representation. Overlapping Max Pool layers are like Max Pool layers with the exception of the adjacent windows over which the maximum determined overlaps, as shown in Figure 5. AlexNet's employed pooling windows with a size of 33 and a stride of 2 between adjacent windows [23]. Figure 5 shows AlexNet architecture.

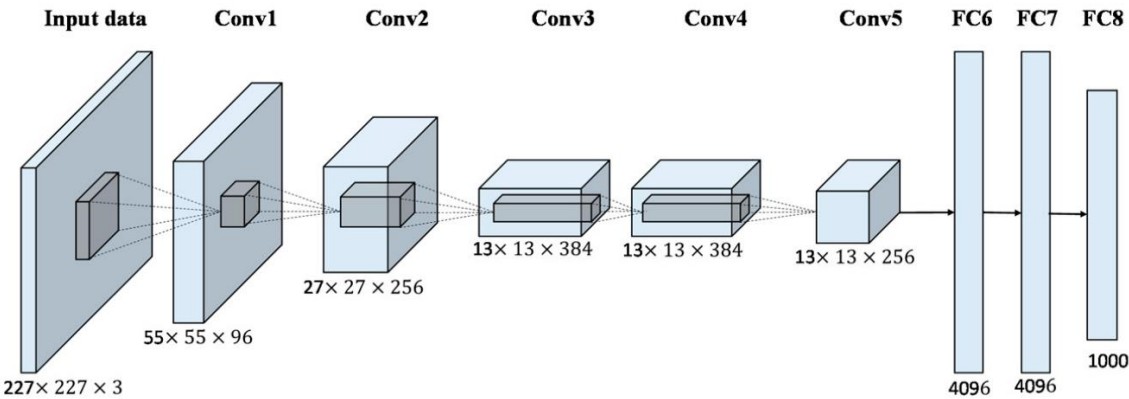

**Figure 5.** AlexNet architecture.

### 3.2.2. VGG16

The use of very small 3 × 3 receptive field (filters) over the whole network with a stride of 1 pixel was proposed in this model. In AlexNet, the receptive field in the first layer was 11 × 11 with stride 4, whereas the receptive field in the second layer was 11 × 11 with stride 4. The notion of utilizing 3 × 3 filters in a uniform manner increased the VGG performance. Two 3 × 3 filters in succession produce a 5 × 5 effective receptive field. Three 3 × 3 filters, meanwhile, produce a 7 × 7 receptive field. A combination of multiple 3 × 3 filters can thus stand in for a larger receptive area. In addition to the three convolution layers, there are three non-linear activation layers, rather than the single

one in 7 × 7. Therefore, the decision functions become more discriminative and would provide the network the ability to converge quickly. Secondly, it greatly reduces the number of weight parameters in the model. This can also be viewed as a regularization of the 7 × 7 convolutional filters, forcing them to decompose through the 3 × 3 filters, with non-linearity introduced in the middle using ReLU activations. The network's inclination to over-fit during the training exercise would be reduced as a result as shown in Figure 6. Furthermore, 3 × 3 is the smallest size that can capture the concepts of left to right, top to bottom, and so on. As a result, lowering the filter size even further may have an influence on the model's capacity to recognize the spatial aspects of the image. The network's persistent use of 3 × 3 convolutions made it incredibly simple, elegant, and easy to deal with [24]. Figure 6 shows VGG16 architecture.

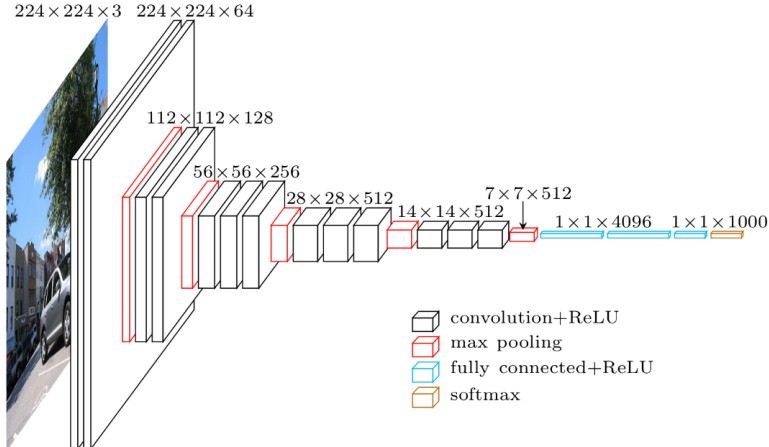

**Figure 6.** VGG16 Architecture.

### 3.2.3. ResNet

We usually stack additional layers in Deep Neural Networks to address a complex problem, which improves accuracy and performance. The idea behind adding more layers is that these layers can learn increasingly complicated features as time goes on. However, it has been discovered that the classic Convolutional neural network model has a maximum depth threshold. Moreover, the emergence of ResNet or residual networks, which are made up of Residual Blocks, has relieved the challenge of training very deep networks. ResNet's skip connections alleviate the problem of disappearing gradients in deep neural networks by allowing the gradient to flow through an additional shortcut channel as shown in Figure 6. These connections also aid the model by allowing it to learn the identity functions, ensuring that the higher layer performs at least as well as the lower layer, if not better [25]. Figure 7 shows ResNet architecture.

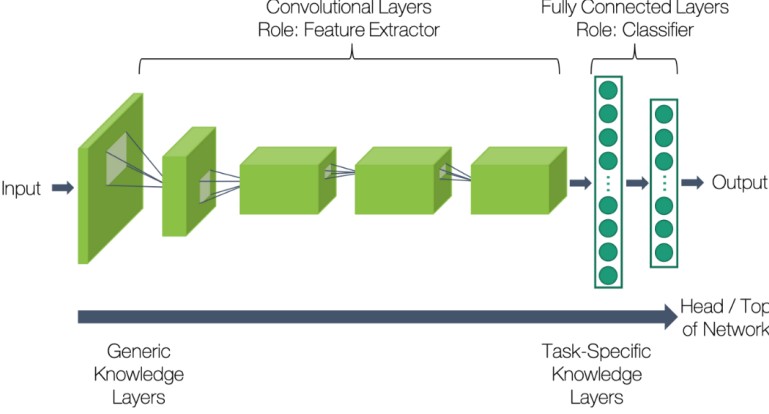

**Figure 7.** ResNet Architecture.

### 3.2.4. EfficientNet

EfficientNet is a convolutional neural network design and scaling method that uses a compound coefficient to scale all depth/width/resolution dimensions evenly. The Efficient-Net scaling method consistently scales network breadth, depth, and resolution with a set of predefined scaling coefficients, unlike traditional methods, which arbitrary scales these elements. Before the EfficientNets, the most popular technique to scale up ConvNets was to increase the depth (number of layers), the breadth (number of channels), or the image quality. As shown in Figure 8, the EfficientNet started by creating a baseline network using a technique called neural architecture search which automates the building of neural networks. On a floating-point operations per second (FLOPS) basis, it optimizes both accuracy and efficiency. The movable inverted bottleneck convolution is used in this architecture (MBConv). The researchers then scaled up this baseline network to create the EfficientNets family of deep learning models [20]. Figure 8 shows EfficientNets architecture.

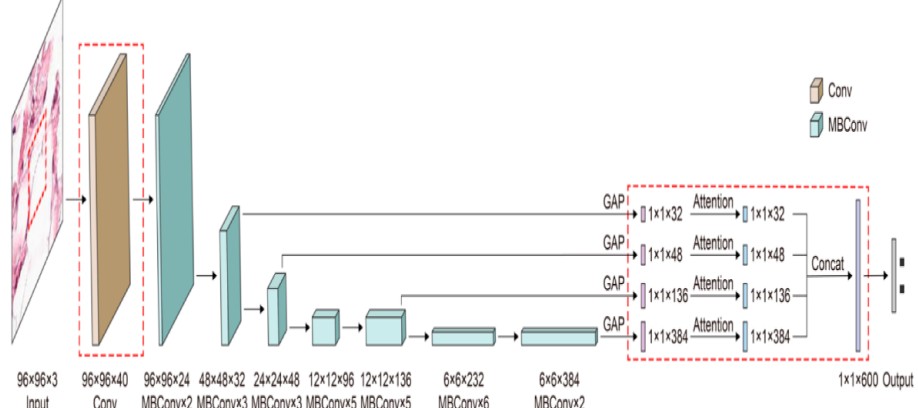

**Figure 8.** Efficient Nets architecture.

In the previous step, the most popular CNN models, according to the literature, were discussed. The next step illustrates how the best CNN model among the four explained CNN models was nominated. In the following, the third step of our methodology is presented.

### 3.3. Select the Best CNN Architecture

According to [26], accuracy is the best benchmark to compare between CNN architectures. Accordingly, experiments have been conducted for testing accuracy among the four selected CNN architectures. Experiments were conducted according to the following steps:

(1) Preparing and resizing the dataset images to be ready for insertion into the selected four CNN architectures.
(2) Labeling the dataset images to be ready for the classification process. The Appendix A shows a snapshot of the software code for labeling the dataset images and for preparing the dataset for experiments.
(3) Split dataset images in two sets, training and testing sets.

The first step in building the real-time object detection model was to split the dataset into train and test sets. The python library split-folders were used to split the dataset. The dataset was split into 80% training set (43,240 images) and 20% test set (10,809 images). The training set was used for training the CNN architecture to achieve the best weights. The testing set was used for checking the correctness of achieved weights and to adjust it accordingly.

Define the optimizer where the Adam optimizer [27] has been chosen for controlling the learning rate based on the following equation:

Optimizer = Adam (lr = 0.001, beta_1 = 0.9, beta_2 = 0.999)

(4) Define the epochs and batch size where the epoch specifies the number of times the CNN accepts whole training data, i.e., the epoch is equal to one forward pass and one backward pass for all training samples. On the other hand, the batch size specifies the number of training samples we use in one forward pass and one backward pass [28]. The following parameters are defined in this step: epochs = 30; batch_size = 32. Due to limitation of this paper size, only the results of last five epoch are presented. In this step, image Augmentation was also implemented to avoid overfitting problem.

(5) Run the four CNN architectures where categorical cross-entropy loss [29] is typically used in a multiclass classification setting in which the outputs are interpreted as predictions of class membership probabilities. The output of this experiment step is defining classification accuracy for each CNN architecture. Table 1 shows the hyperparameters for experiments. Table 2 shows the accuracy results that have been generated from the conducted experiments.

**Table 1.** The Hyperparameters for experiments.

| Hyperparameters | VGG16 Value | RestNet50 Value | EffecientNet Value | AlexNet Value |
|---|---|---|---|---|
| Initial learning rate | 0.001 | 0.001 | 0.001 | 0.001 |
| Activation function | Categorical Cross Entropy | Categorical Cross Entropy | Categorical Cross Entropy | Categorical Cross Entropy |
| Number of epochs | 30 | 30 | 30 | 30 |
| Batch size | 32 | 32 | 32 | 32 |
| Optimizer | ADAM | ADAM | ADAM | ADAM |
| Weight initialization | Xavier initialization | Xavier initialization | Xavier initialization | Xavier initialization |
| Learning rate decay ($\lambda$) | 0.0002 | 0.0002 | 0.0002 | 0.0002 |
| Momentum | 0.9 | 0.9 | 0.9 | 0.9 |

**Table 2.** The accuracy results that have been generated from the conducted experiments.

| Model | Epoch | Train Loss | Train Accuracy (%) | Valid Loss | Valid Acc (%) |
|---|---|---|---|---|---|
| VGG16 | 26 | 0.2187 | 93.27 | 0.5936 | 82.30 |
| | 27 | 0.2069 | 93.70 | | |
| | 28 | 0.2077 | 93.68 | | |
| | 29 | 0.1996 | 93.77 | | |
| | 30 | 0.1957 | 94.05 | | |
| RestNet50 | 26 | 0.0235 | 99.29 | 0.5688 | 89.83 |
| | 27 | 0.0157 | 99.55 | | |
| | 28 | 0.0212 | 99.35 | | |
| | 29 | 0.0175 | 99.52 | | |
| | 30 | 0.1021 | 96.76 | | |
| EffecientNet | 26 | 0.0290 | 99.24 | 0.5481 | 86.56 |
| | 27 | 0.0269 | 99.24 | | |
| | 28 | 0.0225 | 99.32 | | |
| | 29 | 0.0265 | 99.29 | | |
| | 30 | 0.0239 | 99.37 | | |
| AlexNet | 26 | 0.0269 | 99.10 | 0.2004 | 94.81 |
| | 27 | 0.0242 | 99.28 | | |
| | 28 | 0.0265 | 99.19 | | |
| | 29 | 0.0216 | 99.32 | | |
| | 30 | 0.0085 | 99.75 | | |

It is obvious that the best CNN architecture is the AlexNet architecture. In the following table, the implementation of real time recognition model based AlexNet architecture is presented. Figure 9 shows the accuracy results of the conducted experiment. Figure 10 shows the loss results of the conducted experiment. Table 3 shows how many images representing each sign were employed into network training and validation.

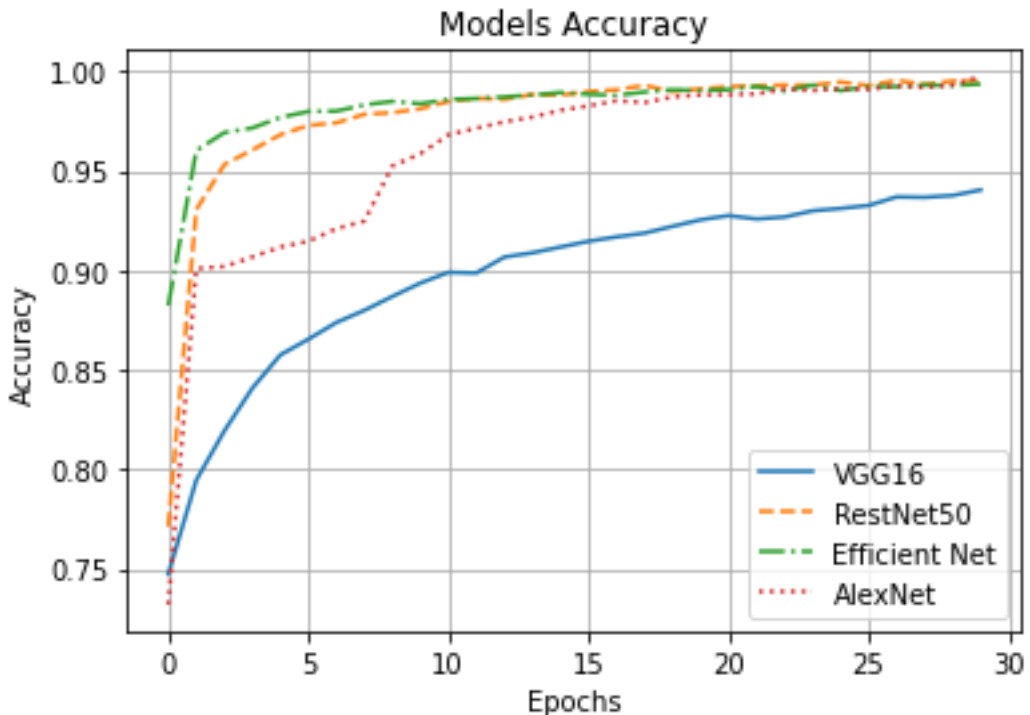

**Figure 9.** The accuracy results of the conducted experiment.

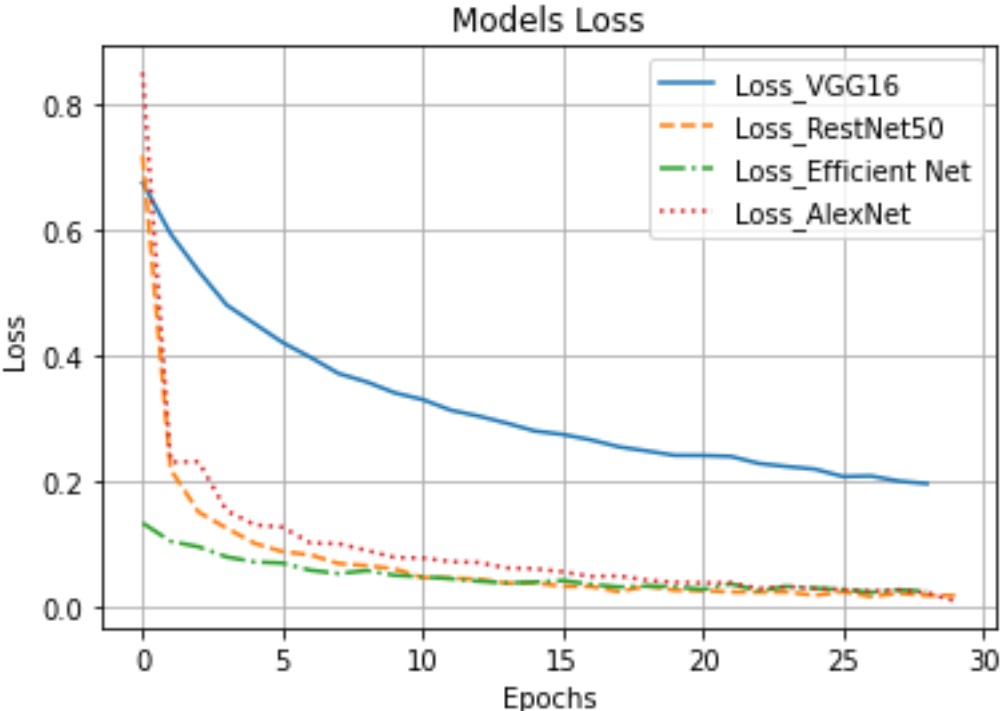

**Figure 10.** The loss results of the conducted experiment.

**Table 3.** How many images representing each sign were employed into network training and validation.

| # | Letter Name in English Script | Letter Name in Arabic Script | # of Images | # | Letter Name in English Script | Letter Name in Arabic Script | # of Images |
|---|---|---|---|---|---|---|---|
| 1 | Alif | (أَلِف)أ | 1672 | 17 | Zā | (ظَاء)ظ | 1723 |
| 2 | Bā | (بَاء) ب | 1791 | 18 | Ayn | (عَين)ع | 2114 |
| 3 | Tā | (أتَاء) ت | 1838 | 19 | Ghayn | (غَين)غ | 1977 |
| 4 | Thā | (ثَاء) ث | 1766 | 20 | Fā | (فَاء)ف | 1955 |
| 5 | Jīm | (جِيمْ) ج | 1552 | 21 | Qāf | (قَاف) ق | 1705 |
| 6 | Hā | (حَاء) ح | 1526 | 22 | Kāf | (كَاف)ك | 1774 |
| 7 | Khā | (خَاء) خ | 1607 | 23 | Lām | (لاَمْ)ل | 1832 |
| 8 | Dāl | (دَألْ) د | 1634 | 24 | Mīm | (مِيمْ)م | 1765 |
| 9 | Dhāl | (ذَال) ذ | 1582 | 25 | Nūn | (نُون)ن | 1819 |
| 10 | Rā | (رَاء) ر | 1659 | 26 | Hā | (هَاء)ه | 1592 |
| 11 | Zāy | (زَاي) ز | 1374 | 27 | Wāw | (وَاو)و | 1371 |
| 12 | Sīn | (سِينْ) س | 1638 | 28 | Yā | (يَا) ئ | 1722 |
| 13 | Shīn | (شِينْ) ش | 1507 | 29 | Tāa | (ة)ة | 1791 |
| 14 | Sād | (صَادْ)ص | 1895 | 30 | Al | (ال)ال | 1343 |
| 15 | Dād | (ضَاد)ض | 1670 | 31 | Laa | (لا)لا | 1746 |
| 16 | Tā | (ظَاء)ط | 1816 | 32 | Yāa | (يَاء) يَاء | 1293 |

## 4. Developing Real Time ArSLA Recognition Model Using AlexNet Deep Learning Architecture

In this section, the technical details of developing a real time model for ArSL recognition is explained. We have selected AlexNet architecture due to its high performance as examined in the previous section. As previously presented in this discussion, the recognition model based on AlexNet architecture has been developed and is ready to be tested and used. Now, Figures 11 and 12 show two Arabic letters Nuon (ن) and Thaa (ث) that have been captured in real time. A gestures technique has been used for identifying and capturing hand shape. In image processing, a gesture is defined as a technique in which part of the human body is recognized by using a camera [30].

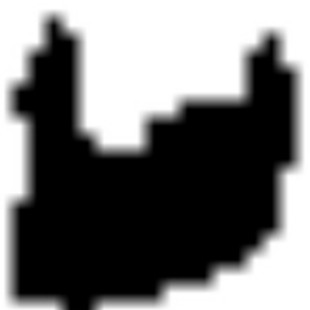

**Figure 11.** Arabic letter Nuon (ن).

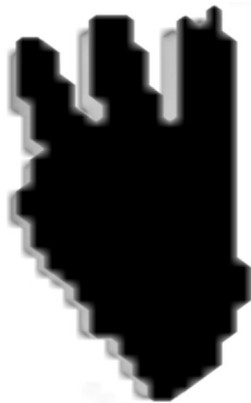

**Figure 12.** Arabic letter Thaa (ث).

In the following, steps for the sign reorganization are described:

Image Capturing: Open-cv has been used for developing software to control the camera and implement the real-time detection. The saved model from previous trainings were loaded into the system for applying with a real-time detector. After that, the gesture recognition model has been used to detect the convexity of hand.

Extracting the ROI: (Region of interest) from inserted frames within background subtraction. Determine the contour and draw the convex hull. The contour is outlined as the object's (hand) boundary that can be seen in the image. The contour can also be a wave connecting points that has a similar color value and is important in the shape analyzing and objects identification method.

Find the convexity defects depending upon the number of defects and determine the gesture. This process takes few milliseconds which means the recognition is implemented at real time.

Algorithm 1 shows the algorithm of gestures technique that was implemented in this model. Algorithm 2 shows the algorithm of testing the model by a single image. The model recognized these two letters successfully. This process can later be used to write a full article. Writing a full article by using our proposed model will enhance the lives of people with hearing disabilities.

---

**Algorithm 1.** Algorithm of gestures technique that has been implemented in this model

---

```
# video capture
Set: cap = cv2.VideoCapture(0)
while True:
    read frame
        # Simulating mirror image
        frame = cv2.flip(frame, 1)
      # Coordinates of the ROI
      x1 = int(0.5*frame.shape [1])
      y1 = 10
      x2 = frame.shape [1]-10
      y2 = int(0.5*frame.shape [1])
      # Drawing the ROI
      # The increment/decrement by 1 is to compensate for the bounding box
      cv2.rectangle(frame, (x1 − 1, y1 − 1), (x2 + 1, y2 + 1), (255, 0, 0), 1)
      # Extracting the ROI
      roi = frame [y1:y2, x1:x2]
 # Resizing the ROI so it can be fed to the model for prediction
      roi = resize image (roi, (64, 64))
      roi = Color CTV (roi, cv2.COLOR_BGR2GRAY)
      _, image = cv2.threshold(roi, 120, 255, cv2.THRESH_BINARY)
      Show image(image)
```

---

**Algorithm 2.** Algorithm of testing the model by single image

```
Set path:
Img = convert to gray(path)
print(path)
letter = path.split(letter)
print(letter)
num = transform(letter)
print(num)
img = reshape image(1, 227, 227, 3)
pred_y = model1.predict(img)
print(np.argmax(pred_y))
calculate test loss
test accuracy = model1.evaluate(img, num)
print(test accuracy)
```

## 5. Discussion and Conclusions

Regarding the confusion matrix, all the models predicted 32 classes for the 32 standard Alphabetic Arabic signs. The VGG model predicted 45 correct images belonging to class 7. It also predicted 30 images correctly belonging to class 19. The ResNet model predicted 20 correct images from class 7 and about 10 images from class 19. The efficientNet model predicted about 10 images correct for class 7 and class 19.

Deep learning challenges were stated in [31]. In the following, the explanation of how this proposal model dealt with these challenges is presented.

Challenge 1: Convolutional neural networks (CNNs) require sufficient training samples to achieve high performance as using small datasets can result in overfitting.

To overcome this challenge, transfer learning has been proposed as a solution, a technique in which the model is trained on a large training set. The results of this training are then treated as the starting point for the target task. Transfer learning has been successful in fields such as language processing and computer vision. Frequently used pretrained deep learning models include common models such as AlexNet, EfficientNet, VGG16, and ResNet, which are all typically utilized for image classification. Data augmentation is another technique that has proved to be effective in alleviating overfitting and improving overall performance. This method increases the training set's size by performing geometric and color transformations such as rotation, resizing, cropping, and either adding noise or blurring the image, etc. In this work, we did not use either transfer learning or data augmentation in training our CNN model. Instead, we utilized the ArSLA dataset which is suitable for testing and training. It uses around 1000 images for each letter to train the CNN model.

Challenge 2: Selecting the proper CNN model. This is because model selection will be different from one dataset to another, meaning the process is dependent on trial and error.

To address this issue, we trained and tested four, state-of-the-art CNN models, including AlexNet, VGG16, GoogleNet, and ResNet, in order to select the best model to classify the sign language. Our results found that AlexNet had the highest accuracy at 94.81%.

Challenge 3: In manufacturing, data acquisition can be difficult. This is due to one of two issues: The first is that sensor placement is ineffective. The second is that vibrations or noise render the collected data useless.

To address this challenge, the ArSLA dataset [6] utilized data from 40 contributors across varying age groups. We then scaled the image pixels to zero-mean and unit variance as pre-processing techniques. We desire our model to learn from noisy images because most of the applications use the camera to capture the letters, some of which are lower quality.

In Section 2 (Related Works), the problem descriptions, research gaps, and enhancement possibilities were discussed in four points. In the following, the contributions of the proposed recognition model are discussed in light of these four points.

(1)　The proposed model is accessible to everyone as no additional or special equipment was used and only a PC or laptop camera is required.

(2)　The process of feature extraction is built into the Alexnet architecture which produces high accuracy. This feature extraction is completely independent of human intervention which in turn makes it free from human error factors.

(3)　The recognition systems developed based on deep learning architectures exhibit a good reputation and produce results with high accuracy. In light of this, the best deep learning (CNN) architectures (according to the literature) were tested by a real and trusted ArSL dataset, then based on the test's results the best CNN architecture were chosen for developing the real-time recognition system. The proposed model was not built from scratch, but rather from the latest findings of researchers in this field, which enabled it to benefit from the accumulated experiences. Table 4 summarizes related works by focusing on training accuracy. In our proposed model, the training accuracy for AlexNet is 99.75% (see Table 2). For the training accuracy that is better than the highest value in related works, see Table 4. For the sake of transparency, we have additionally presented the validation accuracy (testing accuracy) which is 94.81%.

**Table 4.** Shows summary of related works by focus on training accuracy.

| # | Ref | Year | Device | Language | Features | Technique | Training Accuracy |
|---|-----|------|--------|----------|----------|-----------|-------------------|
| 1 | [9] | 2017 | Camera | 25 Arabic words | Image pixels | CNN | 90% |
| 2 | [12] | 2019 | dual Leap Motion Controllers | 100 Arabic words | N geometric parameters | LDA | 88% |
| 3 | [14] | 2019 | Kinect sensor | 35 Indian sign | Distances, angles, and velocity involving upper body joints | Multi-class support vector machine classifier | 87.6% |
| 4 | [11] | 2018 | Single camera | 30 Arabic words | Segmented image | Euclidean distance classifier | 83% |
| 5 | [32] | 2020 | Single camera | 24 English letters | Image pixels | Inception v3 plus Support Vector Machine (SVM) | 92.21% |
| 6 | [16] | 2020 | Single camera | 28 Arabic letters | Image pixels | CNN | 97.82% |
| 7 | [6] | 2015 | Glove | 30 Arabic letters | invariant features | ResNet-18 | 93.4% |
| 8 | [33] | 2011 | Single camera | 20 Arabic words | Edge detection and contours tracking | HMM | 82.22% |
| 9 | [13] | 2019 | Camera | 40 Arabic sign language words | Thresholder image differences | HMM | 94.5% |
| 10 | [10] | 2018 | Camera | 30 Arabic letters | FFT | HOG and SVM | 63.5% |

(4)　The proposed model was developed with the complete Arabic alphabet which allows users to write full articles using ArSLA in real time.

This model is limited to detect only one object (a hand) without taking into the background into consideration. The background of the hand plays a prominent role in object recognition. The performance might not be the same if the background is changed. The background should be the same as in the training set. In addition, the detection process in our proposed model is highly sensitive to variations in the hand's pose.

Future work is planned and suggested to develop a mobile application based on this proposed model. Another direction of future work is to utilize transfer learning to pre-train a model on other sign language datasets such as the American Sign Language dataset, MS-ASL. Other work will modify Arabic sign language data to validate the effectiveness of implementing transfer learning in sign language recognition. Data augmentation will also be applied to generate training samples.

**Author Contributions:** Conceptualization, A.O.E.; Data curation, Z.A.; Formal analysis, S.A.; Investigation, Z.A. and S.A.; Methodology, Z.A. and A.O.E.; Project administration, A.O.E.; Resources, E.A., M.A. and A.A.D.A.; Software, Z.A., E.A., M.A. and A.A.D.A.; Supervision, A.O.E. All authors have read and agreed to the published version of the manuscript.

**Funding:** This research received no external funding.

**Institutional Review Board Statement:** Not applicable.

**Informed Consent Statement:** Not applicable.

**Data Availability Statement:** All data has been presented in main text.

**Conflicts of Interest:** The authors declare no conflict of interest.

## Appendix A

```
# importing libraries
import numpy as np
import pandas as pd
import matplotlib.pyplot as plt
import tensorflow
from tensorflow.keras.layers import Input, Lambda, Dense, Flatten, GlobalAveragePooling2D
from tensorflow.keras.models import Model
from tensorflow.keras.applications.vgg16 import VGG16
from tensorflow.keras.applications import ResNet50
from tensorflow.keras.applications import Xception
from tensorflow.keras.models import Sequential
from tensorflow.keras.applications.vgg16 import preprocess_input
from tensorflow.keras.preprocessing import image
from tensorflow.keras.preprocessing.image import ImageDataGenerator
from tensorflow.keras.models import Sequential
from glob import glob
from tensorflow.keras.applications import EfficientNetB0
# initiate vgg16
model = VGG16(include_top = False, weights = 'imagenet')
x = model.output
x = GlobalAveragePooling2D ( ) (x)
x = Dense(1024, activation = 'relu') (x)
pred = Dense(trainx.num_classes, activation = 'softmax')(x)
mdl2 = Model(inputs = model.input, outputs = pred)
for layer in model.layers:
    layer.trainable = False
mdl2.compile(loss = 'categorical_crossentropy',
            optimizer = 'adam',
              metrics = ['accuracy']                )
mdl2.fit(trainx, epochs = 30)
# vgg model evaluation
test_loss, test_acc = mdl2.evaluate(testx)
# generating predictions
true_classes = testx.classes
class_indices = trainx.class_indices
class_indices = dict((v, k) for k, v in class_indices.items())
ypred2 = mdl2.predict(testx)
ypred2_classes = np.argmax(ypred2, axis = 1)
# accuracy
from sklearn.metrics import accuracy_score
acc = accuracy_score(true_classes, ypred2_classes)
```

```
print(acc)
import seaborn as sns
from sklearn.metrics import confusion_matrix
cm = confusion_matrix(true_classes, ypred2_classes)
print(cm)
plt.figure(figsize = (12, 9))
sns.heatmap(cm)
plt.show()
# RestNet50
res = ResNet50(include_top = False, weights = 'imagenet')
x1 = res.output
x1 = GlobalAveragePooling2D()(x1)
x1 = Dense(1024, activation = 'relu')(x1)
pred1 = Dense(trainx.num_classes, activation = 'softmax')(x1)
mdl3 = Model(inputs = res.input, outputs = pred1)
for layer in model.layers:
    layer.trainable = False
mdl3.compile(loss = 'categorical_crossentropy',
            optimizer = 'adam',
                metrics = ['accuracy']                )
mdl3.fit(trainx, epochs = 30)
# restnet model evaluation
testr_loss, testr_acc = mdl3.evaluate(testx)
# confusion matrix
cm1 = confusion_matrix(trueclas, ypred3_classes)
# heatmap
plt.figure(figsize = (12, 9))
sns.heatmap(cm1)
plt.show()
# efficient net model
eff = EfficientNetB0(include_top = False, weights = 'imagenet')
x2 = eff.output
x2 = GlobalAveragePooling2D()(x2)
x2 = Dense(1024, activation = 'relu')(x2)
pred2 = Dense(trainx.num_classes, activation = 'softmax')(x2)
mdl4 = Model(inputs = eff.input, outputs = pred2)
for layer in model.layers:
    layer.trainable = False
mdl4.compile(loss = 'categorical_crossentropy',
            optimizer = 'adam',
                metrics = ['accuracy']                )
mdl4.fit(trainx, epochs = 30)
# generating predictions for restnet
trueclas = testx.classes
classindices = trainx.class_indices
classindices = dict((v, k) for k, v in class_indices.items())
ypred3 = mdl3.predict(testx)
ypred3_classes = np.argmax(ypred3, axis = 1)
# accuracy
acc1 = accuracy_score(trueclas, ypred3_classes)
print(acc1)
# efficientnet model evaluation
testf_loss, testf_acc = mdl4.evaluate(testx)
# efficientnet confusion matrix
```

```
# confusion matrix
cm2 = confusion_matrix(trueclazz, ypred4_classes)
# heatmap
plt.figure(figsize = (12, 9))
sns.heatmap(cm2)
plt.show()
# Alexnet model
# defining a model
alx = Xception(include_top = False, weights = 'imagenet')
x4 = alx.output
x4 = GlobalAveragePooling2D()(x4)
x4 = Dense(1024, activation = 'relu')(x4)
pred5 = Dense(trainx.num_classes, activation = 'softmax')(x4)
mdl5 = Model(inputs = alx.input, outputs = pred5)

for layer in model.layers:
    layer.trainable = False
mdl5.compile(loss = 'categorical_crossentropy',
            optimizer = 'adam',
            metrics = ['accuracy']            )
mdl5.fit(trainx, epochs = 30)
# alexnet model evaluation
testa_loss, testa_acc = mdl5.evaluate(testx)
# generating predictions for efficientnet
trueclazz1 = testx.classes
clazzindices1 = trainx.class_indices
clazzindices1 = dict((v, k) for k, v in class_indices.items())
ypred5 = mdl5.predict(testx)
ypred5_classes = np.argmax(ypred5, axis = 1)
# accuracy
acc3 = accuracy_score(trueclazz1, ypred5_classes)
print(acc3)

# Alexnet confusion matrix
# confusion matrix
cm3 = confusion_matrix(trueclazz1, ypred5_classes)
# heatmap
plt.figure(figsize = (12, 9))
sns.heatmap(cm3)
plt.show()
```

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
