# Peer review of "A Real Time Arabic Sign Language Alphabets (ArSLA) Recognition Model Using Deep Learning Architecture"

_computers, doi:10.3390/computers11050078_

Round 1
Reviewer 1 Report
This is a study on using deep learning for classification applied to sign language recognition. Here are my comments:
- please clearly explain the way you split your data and the size of the train and test data (showing code in a box is not enough)
- please clearly show in text, the hyperparameters for each model
- it is not necessary but even sometimes confusing to show the code in the manuscript in the form of an image. If you want, you may put them as an appendix. The information shown in the codes should be clearly written in the text so everyone can understand it.
- There is a font problem in Table 9
- you can enrich your discussion by talking about the confusion matrix and telling which signs are most challenging to be recognized.
- despite the opportunities, there are challenges associated with deep models. Please refer to section 6 of "https://doi.org/10.1007/s00170-021-07325-7" where the limitations of deep models are discussed. Can you elaborate a discussion on that and explain the challenges of your model? and possible sources of error?
- referring to the above comment, please combine it with the direction for future research and form a short section at the end of your paper.
Author Response
First, we would like to thank our anonymous reviewer for the valuable comments that have assisted us to improve our work.
- Comment 1: please clearly explain the way you split your data and the size of the train and test data (showing code in a box is not enough)
Answer
The first step to building the real-time object detection model was to split the dataset into train and test sets. The python library split-folders were used to split the dataset. The data set was split into 80% training set (43240 imeges) and 20% test set (10809 imeges )
In the new version, in page 17, the above paragraph has been added to the new version.
- Comment 2: please clearly show in text, the hyperparameters for each model. It is not necessary but even sometimes confusing to show the code in the manuscript in the form of an image. If you want, you may put them as an appendix. The information shown in the codes should be clearly written in the text so everyone can understand it.
Answer: The code has been omitted from the paper and presented in the appendix.
The hyperparameters for the experiments has been presented in Table 1. Table 1 has been added to the new version.
|
AlexNet Value |
EffecientNet Value |
RestNet50 Value |
VGG16 Value |
Hyperparameters |
|
0.001 |
0.001 |
0.001 |
0.001 |
Initial learning rate
|
|
Categorical Cross Entropy
|
Categorical Cross Entropy
|
Categorical Cross Entropy
|
Categorical Cross Entropy
|
Activation function |
|
30 |
30 |
30 |
30 |
Number of epochs
|
|
32 |
32 |
32 |
32 |
Batch size
|
|
ADAM |
ADAM |
ADAM |
ADAM |
Optimizer
|
|
Xavier initialization |
Xavier initialization |
Xavier initialization |
Xavier initialization |
Weight initialization |
|
128 |
128 |
128 |
128 |
Learning rate decay (λ) |
|
0.9 |
0.9 |
0.9 |
0.9 |
Momentum |
- Comment 3: There is a font problem in Table 9
Answer: This problem has been addressed in the new version
- Comment 4: You can enrich your discussion by talking about the confusion matrix and telling which signs are most challenging to be recognized.
Answer: Regarding the confusion matrix, all the models were predicting 32 classes for the 32 standard Alphabetic Arabic signs. The VGG model predicted 45 correct images belonging to class 7. It also predicts 30 images correctly belonging to class 19. The ResNet model predicted 20 correct images of class 7 and about 10 images for class 19. The efficientNet model predicted about 10 images correct for class 7 and class 19.
In the new version, the above answer has been added to the conclusion and discussion section as has been suggested.
- Comment 5: despite the opportunities, there are challenges associated with deep models. Please refer to section 6 of "https://doi.org/10.1007/s00170-021-07325-7" where the limitations of deep models are discussed. Can you elaborate a discussion on that and explain the challenges of your model? and possible sources of error?
Answer
Deep learning challenges have been stated in Nasir and Sassani (2021), in the following, the explanation of how this proposal model has dealt with these challenges has been presented.
Challenge 1: Convolutional neural networks (CNNs) require sufficient training samples to achieve high performance, as using small datasets can result in overfitting.
To overcome this challenge Transfer learning has been proposed as solution, a technique in which the model is trained on a large training set. The results of this training are then treated as the starting point for the target task. Transfer learning has been successful in fields such as language processing and computer vision. Frequently used pretrained deep learning models such common models, including AlexNet, EfficientNet, VGG16, and ResNet, are typically utilized for image classification. Data augmentation is another technique that also been effective in alleviating overfitting and improving overall performance. This method increases the training set’s size by performing geometric and color transformations such as rotation, resizing, cropping, and adding noise to or blurring the image, etc. In this work, we did not use either transfer learning or data augmentation in training our CNN model. Instead, we utilized the ArSLA dataset, which is suitable for testing and training. It uses around 1000 image for each letter to train the CNN model.
Challenge 2: Selecting the proper CNN model. This is because model selection will be different from one dataset to another, meaning the process is dependent on trial and error.
To address this issue, we trained and tested four, state-of-the-art CNN models, including AlexNet, VGG16, GoogleNet, and ResNet, to the select the best model for classifying the sign language. Our results found that AlexNet had the highest accuracy, at 96%.
Challenge 3: In manufacturing, data acquisition can be difficult. This is due to one of two issues: The first is that sensor placement is ineffective. The second is that vibrations or noise render the collected data useless.
To address this, ArSLA data set (Tharwat et al., 2015) utilized data from 40 contributors, across varying age groups. We then scaled the image pixels to zero-mean and unit variance as pre-processing techniques. We are interested in having our model learn from noisy images because most of the applications use the camera to capture the letters, some of which are lower quality.
In the new version, the above answer has been added to the conclusion and discussion section as has been suggested.
- Comment 6: referring to the above comment, please combine it with the direction for future research and form a short section at the end of your paper.
Answer: In future work, transfer learning will be utilized to pre-train a model on other sign language datasets such as American Sign Language dataset MS-ASL. Other work will modify Arabic sign language data to validate the effectiveness of implementing transfer learning in sign language recognition. Data augmentation will also be applied to generate training samples.
In the new version, the above answer has been added to the conclusion and discussion section as has been suggested.

Reviewer 2 Report
>I suppose, there are typos in the abstract.
>The final sentence presented in the abstract should be rewritten.
>Please introduce additional, more precise, keywords.
>The introduction should contain more information related to the speech recognition techniques, deep learning methods and hardware/software implementations.
>Have you considered other resolutions of the images (section 3.1)?
>Please improve the description of tables (section 3.3).
>The article needs to be edited and reformatted carefully.
>The symbols used in the equations should be described properly.
>The parts of the code, presented in the tables, should be presented as a pseudo-code or a scheme.
>The details of the user-defined training coefficients selection should be described (e.g. section 3.3).
>How were the networks initialized (values of internal parameters before training)?
>How do the neural algorithms work under disturbances (e.g. quality of input images)?
Author Response
First, we would like to thank our anonymous reviewer for the valuable comments that have assisted us to improve our work.
Comment 1: I suppose, there are typos in the abstract. The final sentence presented in the abstract should be rewritten.
Answer: The abstract has been revised and updated accordingly. The final sentence has been updated to become (The model has been developed based on Alexnet architecture and was successfully tested at real time with 94.81% accuracy rate.)
Comment 2: Please introduce additional, more precise, keywords.
Answer: Two additional key words have been added. In the new version, the keywords are: Deep Learning; Arabic Sign Language Alphabetic; AlexNet Architecture; Transfer learning; Data augmentation
Comments 3: The introduction should contain more information related to the speech recognition techniques, deep learning methods and hardware/software implementations.
Answer: This paper has nothing to do with speech recognition.
In the new version, in introduction, the following paragraph has been added
Transfer learning (Zhuang et al., 2020) has been proposed as solution to overcome this challenge, a technique in which the model is trained on a large training set. The results of this training are then treated as the starting point for the target task. Transfer learning has been successful in fields such as language processing and computer vision. Data augmentation (Perez and Wang, 2017) is another technique that also been effective in alleviating overfitting and improving overall performance. This method increases the training set’s size by performing geometric and color transformations such as rotation, resizing, cropping, and adding noise to or blurring the image, etc. In this work, we did not use either transfer learning or data augmentation in training our CNN model. Instead, we utilized the ArSLA dataset, which is suitable for testing and training. It uses around 1000 image for each letter to train the CNN model.
Comment 3: Have you considered other resolutions of the images (section 3.1)?
Answer: The four selected deep learning architectures have been tested under standard ArSLA standard dataset. According to the best of our knowledge, this is the only ArSLA dataset is available.
Comment 4: Please improve the description of tables (section 3.3).
Answer: The tables that show codes have been omitted from the paper and presented in the appendix.
Comment 5: The article needs to be edited and reformatted carefully.
Answer: The new version of the paper has been edited and formatted carefully.
Comment 6: The symbols used in the equations should be described properly.
Answer: In the new version, the symbols that are used in the equations have been described properly
Comment 7: The parts of the code, presented in the tables, should be presented as a pseudo-code or a scheme.
Answer: The code has been omitted from the paper and presented in the appendix.
Comment 8: The details of the user-defined training coefficients selection should be described (e.g. section 3.3).
Answer: The hyperparameters for the experiments has been presented in Table 1. Table 1 has been added to the new version.
|
AlexNet Value |
EffecientNet Value |
RestNet50 Value |
VGG16 Value |
Hyperparameters |
|
0.001 |
0.001 |
0.001 |
0.001 |
Initial learning rate
|
|
Categorical Cross Entropy
|
Categorical Cross Entropy
|
Categorical Cross Entropy
|
Categorical Cross Entropy
|
Activation function |
|
30 |
30 |
30 |
30 |
Number of epochs
|
|
32 |
32 |
32 |
32 |
Batch size
|
|
ADAM |
ADAM |
ADAM |
ADAM |
Optimizer
|
|
Xavier initialization |
Xavier initialization |
Xavier initialization |
Xavier initialization |
Weight initialization |
|
128 |
128 |
128 |
128 |
Learning rate decay (λ) |
|
0.9 |
0.9 |
0.9 |
0.9 |
Momentum |
Comment 9: How were the networks initialized (values of internal parameters before training)?
Answer: Same as the answer for previous comment.
Comment 10: How do the neural algorithms work under disturbances (e.g. quality of input images)?
Answer: We did not test the model with different image qualities. The four selected deep learning architectures have been tested under standard ArSLA standard dataset. According to the best of our knowledge, this is the only ArSLA dataset is available

Reviewer 3 Report
The paper presents a method for Arabic Sign Language Alphabets recognition with use of deep networks. The paper is clearly written, its contribution is to validate various basic deep network architectures for ArSLA recognition. The experiments are carefully planned and correctly performed. Thus, presented conclusions are mostly sound. There are still some issues that should be addressed before the paper will be suitable for publication.
1. It was not clearly stated, how many images representing each sign were employed into network training and validation.
2. The paper lacks discussion section. The obtained results should be compared discussed; it would be valuable to explain why certain DN perform better than others.
3. Obtained results should be also compared with these obtained by the other researchers.
4. The title suggests that proposed solution operates in real time. Computing time of implemented netorks was not provided, however. Please demonstrate, that your solution is “real time” indeed. This means that sign recognition should occur between acquisition of the consecutive images, without delaing the acquisition.
5. Limitation of the proposed approach should be also discussed.
6. I recommend removing the software code fragments from the main text and place them in the appendix. Currently, they make it difficult to follow the paper.
Author Response
First, we would like to thank our anonymous reviewer for the valuable comments that have assisted us to improve our work.
The paper presents a method for Arabic Sign Language Alphabets recognition with use of deep networks. The paper is clearly written, its contribution is to validate various basic deep network architectures for ArSLA recognition. The experiments are carefully planned and correctly performed. Thus, presented conclusions are mostly sound. There are still some issues that should be addressed before the paper will be suitable for publication.
It was not clearly stated, how many images representing each sign were employed into network training and validation.
Answer
In the new version, Table 3 that shows how many images representing each sign were employed into network training and validation has been added.
|
# |
Letter name in English Script |
Letter name in Arabic script |
# of Images |
# |
Letter name in English Script |
Letter name in Arabic script |
# of images |
|
1 |
Alif |
أَلِف)أ) |
1672 |
17 |
Zā |
ظَاء)ظ) |
1723 |
|
2 |
Bā |
بَاء) ب) |
1791 |
18 |
Ayn |
عَين)ع) |
2114 |
|
3 |
Tā |
أتَاء) ت) |
1838 |
19 |
Ghayn |
غَين)غ) |
1977 |
|
4 |
Thā |
ثَاء) ث) |
1766 |
20 |
Fā |
فَاء)ف) |
1955 |
|
5 |
Jīm |
جِيمْ) ج) |
1552 |
21 |
Qāf |
قَاف) ق) |
1705 |
|
6 |
Hā |
حَاء) ح) |
1526 |
22 |
Kāf |
كَاف)ك) |
1774 |
|
7 |
Khā |
خَاء) خ) |
1607 |
23 |
Lām |
لاَمْ)ل) |
1832 |
|
8 |
Dāl |
دَالْ) د) |
1634 |
24 |
Mīm |
مِيمْ)م) |
1765 |
|
9 |
Dhāl |
ذَال) ذ) |
1582 |
25 |
Nūn |
نُون)ن) |
1819 |
|
10 |
Rā |
رَاء) ر) |
1659 |
26 |
Hā |
هَاء)ه) |
1592 |
|
11 |
Zāy |
زَاي) ز) |
1374 |
27 |
Wāw |
وَاو)و) |
1371 |
|
12 |
Sīn |
سِينْ) س) |
1638 |
28 |
Yā |
يَا) ئ) |
1722 |
|
13 |
Shīn |
شِينْ) ش) |
1507 |
29 |
Tāa |
ة)ة) |
1791 |
|
14 |
Sād |
صَادْ)ص) |
1895 |
30 |
Al |
ال)ال) |
1343 |
|
15 |
Dād |
ضَاد)ض) |
1670 |
31 |
Laa |
ﻻ)ﻻ) |
1746 |
|
16 |
Tā |
طَاء)ط) |
1816 |
32 |
Yāa |
يَاء) يَاء) |
1293 |
- The paper lacks discussion section. The obtained results should be compared discussed; it would be valuable to explain why certain DN perform better than others.
Answer
In the new version, the following statements have been added to discussion and conclusion part.
Regarding the confusion matrix, all the models were predicting 32 classes for the 32 standard Alphabetic Arabic signs. The VGG model predicted 45 correct images belonging to class 7. It also predicts 30 images correctly belonging to class 19. The ResNet model predicted 20 correct images of class 7 and about 10 images for class 19. The efficientNet model predicted about 10 images correct for class 7 and class 19.
Deep learning challenges have been stated in Nasir and Sassani (2021), in the following, the explanation of how this proposal model has dealt with these challenges has been presented.
Challenge 1: Convolutional neural networks (CNNs) require sufficient training samples to achieve high performance, as using small datasets can result in overfitting.
To overcome this challenge Transfer learning has been proposed as solution, a technique in which the model is trained on a large training set. The results of this training are then treated as the starting point for the target task. Transfer learning has been successful in fields such as language processing and computer vision. Frequently used pretrained deep learning models such common models, including AlexNet, EfficientNet, VGG16, and ResNet, are typically utilized for image classification. Data augmentation is another technique that also been effective in alleviating overfitting and improving overall performance. This method increases the training set’s size by performing geometric and color transformations such as rotation, resizing, cropping, and adding noise to or blurring the image, etc. In this work, we did not use either transfer learning or data augmentation in training our CNN model. Instead, we utilized the ArSLA dataset, which is suitable for testing and training. It uses around 1000 image for each letter to train the CNN model.
Challenge 2: Selecting the proper CNN model. This is because model selection will be different from one dataset to another, meaning the process is dependent on trial and error.
To address this issue, we trained and tested four, state-of-the-art CNN models, including AlexNet, VGG16, GoogleNet, and ResNet, to the select the best model for classifying the sign language. Our results found that AlexNet had the highest accuracy, at 94.81%.
Challenge 3: In manufacturing, data acquisition can be difficult. This is due to one of two issues: The first is that sensor placement is ineffective. The second is that vibrations or noise render the collected data useless.
To address this challenge, ArSLA data set (Tharwat et al., 2015) utilized data from 40 contributors, across varying age groups. We then scaled the image pixels to zero-mean and unit variance as pre-processing techniques. We are interested in having our model learn from noisy images because most of the applications use the camera to capture the letters, some of which are lower quality.
- Obtained results should be also compared with these obtained by the other researchers.
Answer
In section 2, related works have been discussed and analyzed. Our proposed model has achieved 94.81% accuracy rate with is higher than any related works. In section 2, the research gap has been highlighted in 4 points which later has been used in section 5, Discussion and Conclusion, to prove the contribution of this proposed model.
The title suggests that proposed solution operates in real time. Computing time of implemented networks was not provided, however. Please demonstrate, that your solution is “real time” indeed. This means that sign recognition should occur between acquisition of the consecutive images, without delaing the acquisition.
Answer
In the new version, this paragraph has been added:
In the following, steps for the sign reorganization has been described
Image Capturing: Open-cv has been used for developing software to control the camera and implment real-time detection. The saved model from previous training were loaded into the system for applying real-time detector. After that The gesture recognition gas been detected the convexity of hand.
Extracting the ROI: (Region of interest) from inserted frames withing background subtraction.
Find out the contour draw the convex hull. The contour is outlined as object’s (hand) boundary that can be seen in the image. The contour can also be a wave connecting points that has the similar color value and is important in shape analyzing, objects identification method.
Find the convexity defects depending upon the number of defects and find out the gesture.
This process is taking seconds which makes the recognition is implemented at a real time.
Limitation of the proposed approach should be also discussed.
Answer
The proposed model is limited to the 32 signs of Arabic letters that are included in the used dataset.
I recommend removing the software code fragments from the main text and place them in the appendix. Currently, they make it difficult to follow the paper.
Answer
The tables that show codes have been omitted from the paper and presented in the appendix.

Round 2
Reviewer 1 Report
I appreciate the authors for addressing my comments.
Author Response
We would like to thank our anonymous reviewer again for the valuable comments and questions and for his time and effort for assisting us to improve our work.
Reviewer 2 Report
-
Author Response

(The authors gave the same response as above.)

Reviewer 3 Report
Thank you for your answers, but I'm only partially satisfied.
- I urge you to refer to the results obtained by other authors. This comparison, preferably in the form of a table, should be included in the Discussion section (even if some information was provied in Section 2)
- Please provide the exact time of calculations ("a few seconds" is not a precise definition). The authors' responses show that the method does not work in real time. Real-time operation means that the algorithm does not delay the operation of the hardware used (in this case the camera for image acquisition of the text image). "a few seconds" is much longer than time required for image acquisition. Please remove the information that the algorithm works in real time.
- Again, please provide the limitations of the developed method (are you sure the only limitation is the recognition of 32 characters?)
Author Response
We would like to thank our anonymous reviewer again for the valuable comments and questions and for his time and effort for assisting us to improve our work.
- I urge you to refer to the results obtained by other authors. This comparison, preferably in the form of a table, should be included in the Discussion section (even if some information was provied in Section 2)
Answer
In the new version, we have added Table 6. In page 19, we have added the following sentences:
Table 6 shows summary of related works by focus on training accuracy. In our proposed model, the training accuracy for AlexNet is 99.75% see Table 2 which is better than the highest value in related works, see Table 6. For sake of transparency, we have present in addition the validation accuracy (Testing accuracy) which is 94.81%.
Table 6 shows summary of related works by focus on training accuracy
|
# |
Title |
Year |
Device |
Language |
Features |
Technique |
Training Accuracy |
|
1 |
Ref 5 |
2017 |
Camera |
25 Arabic words |
Image pixels |
CNN |
90% |
|
2 |
Ref 4 |
2019 |
dual Leap Motion Controllers |
100 Arabic words |
N geometric parameters |
LDA |
88% |
|
3 |
Ref 7 |
2019 |
Kinect sensor |
35 Indian sign |
Distances, angles, and velocity involving upper body joints |
Multi-class support vector machine classifier |
87.6% |
|
4 |
Ref 12 |
2018 |
Single camera |
30 Arabic words |
Segmented Image |
Euclidean distance classifier |
83% |
|
5 |
Ref3 |
2020 |
Single camera |
24 English letters
|
Image pixels |
Inception v3 plus Support Vector Machine (SVM) |
92.21% |
|
6 |
Ref 21 |
2020 |
Single camera |
28 Arabic letters |
Image pixels |
CNN |
97.82% |
|
7 |
Ref 29 |
2015 |
Glove |
30 Arabic letters |
invariant features |
ResNet-18 |
93.4% |
|
8 |
Ref 31 |
2011 |
Single camera |
20 Arabic words |
Edge detection and contours tracking |
HMM |
82.22% |
|
9 |
Ref 10 |
2019 |
Camera |
40 Arabic sign language words |
Thresholder image differences |
HMM |
94.5% |
|
10 |
Ref 1 |
2018 |
Camera |
30 Arabic letters |
FFT |
HOG and SVM |
63.5% |
- Please provide the exact time of calculations ("a few seconds" is not a precise definition). The authors' responses show that the method does not work in real time. Real-time operation means that the algorithm does not delay the operation of the hardware used (in this case the camera for image acquisition of the text image). "a few seconds" is much longer than time required for image acquisition. Please remove the information that the algorithm works in real time.
Answer
According to (Pulli, K., Baksheev, A., Kornyakov, K., & Eruhimov, V. (2012). Real-time computer vision with OpenCV. Communications of the ACM, 55(6), 61-69.) if FPS is between(16 to 17 FPS) then the display is in real-time.
In our proposed model the average sign detection speed is 0.1 seconds which is equal to 16-17 FPS.
FPS = frames per second
In the following, the code for claucating FPS
import time
fpsLimit = 1 # throttle limit
startTime = time.time()
cv = cv2.VideoCapture(0)
While True:
frame = cv.read
nowTime = time.time()
if (int(nowTime - startTime)) > fpsLimit:
startTime = time.time() # reset time
- Again, please provide the limitations of the developed method (are you sure the only limitation is the recognition of 32 characters?)
Answer
In the new version, in page 20, we have added this statement:
This model is limited to detect only one object which a hand with take into the consideration the background. Background of the hand plays a prominent role in object recognition. the performance might not the same when we change the background, background should be same as like training set. In addition, the detection process in our proposed model is highly sensitive to pose variations.

Round 3
Reviewer 3 Report
Than you for correctly addressing all issues raised in my review. The paper now is suitable for publication.